# Proportion and seasonality of blood parasites in animals in Mosul using the Veterinary Teaching Hospital Lab data

**Hussam M. S. Alimam**[1], **Dhiyaa A. Moosa**[1☯], **Eva A. Ajaj**[1☯], **Mohammad O. Dahl**[1]*, **Israa A. Al-Robaiee**[1☯], **Semaa F. Hasab Allah**[1,2☯], **Zahraa M. Al-Jumaa**[1☯], **Eman D. Hadi**[1☯]

**1** Department of Internal and Preventive Medicine, College of Veterinary Medicine, University of Mosul, Mosul, Nineveh, Iraq, **2** Office of Vice President for Scientific Affairs, University of Al-Hamdaniya, Bartella, Nineveh, Iraq

☯ These authors contributed equally to this work.
* mdahl@uomosul.edu.iq

**Data Availability Statement:** All relevant data are within the paper.

**Funding:** The author(s) received no specific funding for this work.

## Abstract

Several local studies have examined evidence of blood parasites in different animals in Mosul; however, information about the most prevalent parasite and the seasonality of the infection remains limited. The objective of the study conducted here was to investigate the proportion and seasonality of blood parasites in animals in Mosul using the Veterinary Teaching Hospital Lab data. Laboratory records for a period of 25 months were used for data retrieval. In all included animals, Giemsa-stained blood smears were examined by an attending clinical pathologist for the presence of parasites. Seasons were assigned on a basis of examination date, and the seasonality was quantified by estimating season-to-season ratio. The results indicated that 61.77% of examined animals were tested positive for blood parasites. The most evident parasites were *Trypanosoma* spp., *Theileria* spp., *Babesia* spp., and then *Anaplasma* spp., with evidence of mixed infection. The odds of the infection did not significantly vary in different age groups. There was a marked linear pattern in the seasonality of the infection with *Trypanosoma* spp. and *Anaplasma* spp. An increase of the infection during spring and autumn with *Theileria* spp. and *Babesia* spp. was also evident. In conclusion, infection with blood parasites in different animals in Mosul is common with substantial burden, the effect of age-related infection is negligible, and the seasonality of the infection is evident.

## Introduction

Blood parasites are frequently diagnosed in different animals examined at the University of Mosul's Veterinary Teaching Hospital lab. A recent analysis of the records of this hospital indicated that blood parasitic infections constituted the second most frequent cases diagnosed at the hospital, and the most frequent cases in cattle [1]. In a study conducted on 220 cattle from various areas in Mosul, *Theileria* spp., *Babesia* spp., *Anaplasma* spp., *Mycoplasma wenyonii*, *Trypanosoma* spp., tachyzoites of *Toxoplasma gondii*, Microfilaria of *Setaria* spp., and *Ehrlichia*

**Competing interests:** The authors have declared that no competing interests exist.

spp. were identified in blood smears [2]. Previous studies diagnosed *Anaplasma marginale* in cows [3], *Theileria hirci* in local sheep [4], *Theileria hirci* and *Anaplasma ovis* in local goats [5], *Babesia equi* and *Babesia caballi* in horses [6], and *Babesia caballi* in camels [7].

Seasonality of blood parasite infection in animals is evident as a function of the seasonal prevalence of the vectors [8]. A recent local epidemiological study indicated that the lowest odds of blood parasites infection were observed in winter, and appeared to be the highest during autumn [1]. Indeed, the prevalence of *Trypanosoma* spp. is increased three months after the peak of the dry season when the tsetse fly abundance is peaked [9]. During the dry season, adult tsetse flies rely on the blood from the host to adapt to the dry environment [10]. On the other hand, *Theileria* spp., *Babesia* spp., and *Anaplasma* spp. infections are increased with the increase of tick infestation during the rainy season, then summer, and at the lowest rate during winter [11–13].

Information about the proportion and seasonality of blood parasites infection in animals in Mosul is needed. Despite the several studies concerning this type of infection accomplished locally, information about the most prevalent parasite and the seasonality of the infection remains limited. That is, most previous studies such as those by Al-Obaidi and Alsaad [4], Al-Saad and AL-Mola [6], Alsaad et al. [14], and Al-Badrani [15] have focused on studying the clinical and hematological effects of blood parasites on affected animals rather than epidemiological evidence of the infection. Although the study by Al-Abadi and Al-Badrani [2] quantified the proportion of the infection for several parasites, the seasonality was overlooked. In contrast, the study by Dahl et al. [1] calculated the odds of the infection among different seasons without details for each type of parasite. Therefore, the objective of the study conducted here was to investigate the proportion and seasonality of blood parasites in animals in Mosul using the Veterinary Teaching Hospital Lab data. This hospital is one of the main veterinary clinics in Mosul city and receives animals raised in Nineveh governorate, Iraq, particularly Mosul city and its countryside. Records of this hospital have been identified as a source for studying some epidemiological factors of diseases in animals [1].

## Materials and methods

### Study animals

Animals examined at the University of Mosul Veterinary Teaching Hospital clinical pathology laboratory for evidence of blood parasites in blood smears were considered for inclusion in this study. These animals included cattle, sheep, goats, buffaloes, horses, and dogs. Laboratory records for a period of 25 months (October 30, 2017, to November 30, 2019) were used for data retrieval. Included animals were firstly examined clinically at the hospital's internal medicine section. Animals that exhibited icteric mucous membranes, enlargement of superficial lymph nodes, hemoglobinuria, tick infestations, and/or prolonged illness were usually considered for further examination via blood smear.

### Laboratory examination

Giemsa-stained blood smears were examined under an oil immersion lens by an attending clinical pathologist for the presence of parasites. In this work, different blood parasites were identified as the following (i) *Trypanosoma* spp. were intercellular elongated and flagellated trypomastigotes, (ii) *Theileria* spp. were intraerythrocytic round to oval with a nucleus or elongate with bipolar chromatin bodies trophozoites, (iii) *Babesia* spp. were intraerythrocytic round, ovoid, elongate, or amoeboid trophozoites or pyriform paired merozoites, and (iv) *Anaplasma* spp. were tiny intraerythrocytic spheres [16].

## Data collection

Collected data included: examination date, animal species and age, in addition to results of blood smears. Seasons were assigned on a basis of the examination date. In this study, variables for season and age were identified as previously defined by Dahl et al. [1] as the following: season included winter (1$^{st}$ Dec to 28$^{th}$ Feb); spring (1$^{st}$ March to 31$^{st}$ May); summer (1$^{st}$ Jun to 30$^{th}$ Sept); and autumn (1$^{st}$ Oct to 30$^{th}$ Nov), whereas age included young ($<$ 1-year-old) and adult ($\geq$ 1-year-old). Finally, results of blood smears were identified as positive or negative for *Trypanosoma* spp., *Theileria* spp., *Babesia* spp., and *Anaplasma* spp., and either single or mixed infection.

## Statistical analysis

In this study, the proportion of a particular blood parasite constituted the number of smears tested positive for that parasite to the total number of smears tested positive. The association between the infection and age of the animal was examined via the conditional logistic regression, where the data was matched according to the type of the animal, the odds ratio (OR) was considered to measure the magnitude of the association, and value of $P \leq 0.05$ (two-tailed) was considered significant [17]. On the other hand, the seasonality was measured via a direct comparison between the proportions of a particular parasite infection among different seasons [18]. In addition, the seasonality was quantified by estimating season-to-season ratio taking into consideration differences in the length of each season using an equation adapted from Ferreira et al. [19] as the following: S1-to-S2 ratio = (P1/S1$_{length}$)/(P2/S2$_{length}$), where;

S1 = season one (e.g., spring),

S2 = season two (e.g., summer),

P1 = proportion of a particular parasite detected during season one,

P2 = proportion of a particular parasite detected during season two,

S1$_{length}$ = length of the season one in days,

S2$_{length}$ = length of the season two in days,

## Results

Among 552 blood smears were examined, 341 (61.77%) were tested positive for blood parasites from different animals. Generally, *Trypanosoma* spp. was the most reported blood parasite, followed by *Theileria* spp., *Babesia* spp., and then *Anaplasma* spp. (Fig 1). In this data, *Trypanosoma* spp. was the most evident blood parasite in ruminants (cattle, sheep, goats, and buffalo), *Theileria* spp. was frequently reported in all tested animals except dogs, *Babesia* spp. constituted the most blood parasite diagnosed in blood smears from dogs, and *Anaplasma* spp. was detected in cattle only (Table 1). Mixed infection with at least two types was reported in only 4.5% of the positive cases (Fig 2). Additionally, although the differences in the proportion of the infection by *Trypanosoma* spp. was more than 10% greater in young compared to adult animals (44.87%– 34.39% = 10.48%), the odds did not reach the assigned statistical significance (*P*-value = 0.09; Table 2). On the other hand, the differences in the proportion of infection by *Anaplasma* spp., *Babesia* spp., and *Theileria* spp. among young and adult animals were not significant (*P*-values > 0.05; Table 2).

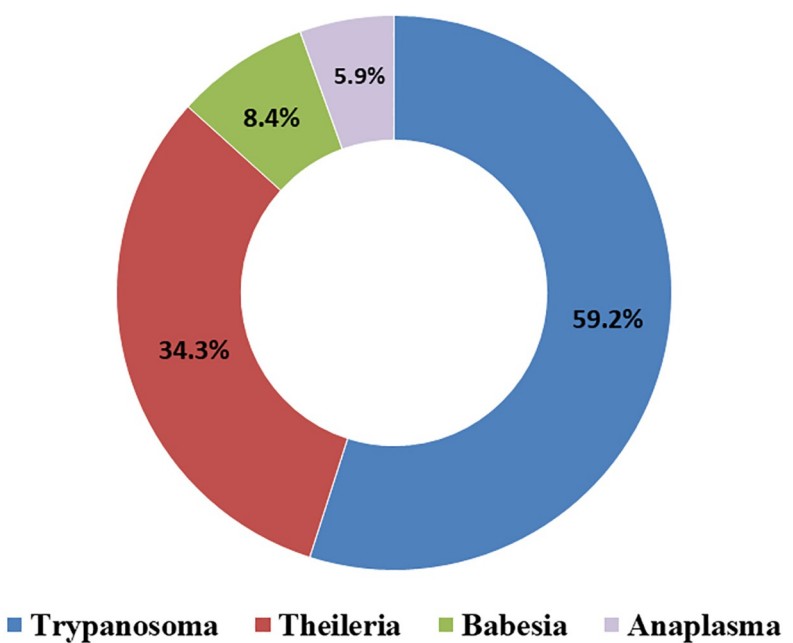

**Fig 1. The proportion of blood parasites detected in blood smears from animals tested at the Veterinary Teaching Hospital | University of Mosul between Oct 30, 2017 to Dec 31, 2019.**

The results revealed that there was an evident pattern in the seasonality of the infection (Fig 3, Table 3). In *Trypanosoma* spp. infection, there was linear increase in the percentage of the infection from spring to autumn to be declined in winter. On the other hand, infections with *Theileria* spp. and *Babesia* spp. slightly increased in spring and autumn compared to summer, while markedly decreased during winter. Finally, there was linear decrease in the percentage of the infection with *Anaplasma* spp. during the year from spring to winter.

## Discussion

The study conducted here revealed that *Trypanosoma* spp. was the most evident blood parasite diagnosed in the animals in Mosul, followed by *Theileria* spp. In addition, *Trypanosoma* spp. was mostly reported during autumn, whereas both *Theileria* spp. and *Babesia* spp. were mostly

**Table 1. Distribution of blood parasites according to the type of animal using blood smears tested at the Veterinary Teaching Hospital | University of Mosul between Oct 30, 2017 to Dec 31, 2019.**

| | Tested Animals | | Trypanosoma | Theileria | Babesia | Anaplasma |
|---|---|---|---|---|---|---|
| Type | N | +ve | n (%) | n (%) | n (%) | n (%) |
| Cattle | 351 | 223 | 128 (57.40) | 83 (37.22) | 10 (4.48) | 20 (8.97) |
| Sheep | 114 | 77 | 49 (63.64) | 23 (29.87) | 9 (11.69) | 0 |
| Goats | 22 | 15 | 8 (53.33) | 5 (33.33) | 2 (13.33) | 0 |
| Buffalo | 28 | 15 | 11 (73.33) | 6 (40) | 1 (6.67) | 0 |
| Horse | 19 | 5 | 1 (20) | 2 (40) | 2 (40) | 0 |
| Dog | 18 | 6 | 1 (16.67) | 0 | 5 (83.33) | 0 |

Abbreviations: (N): total number of tested animals, (+ve): number of animals with blood smear tested positive for at least one type of blood parasite, (n): number of animals tested positive for a particular parasite, and (%): proportion of a particular parasite among positive blood smears for the same type of animal.

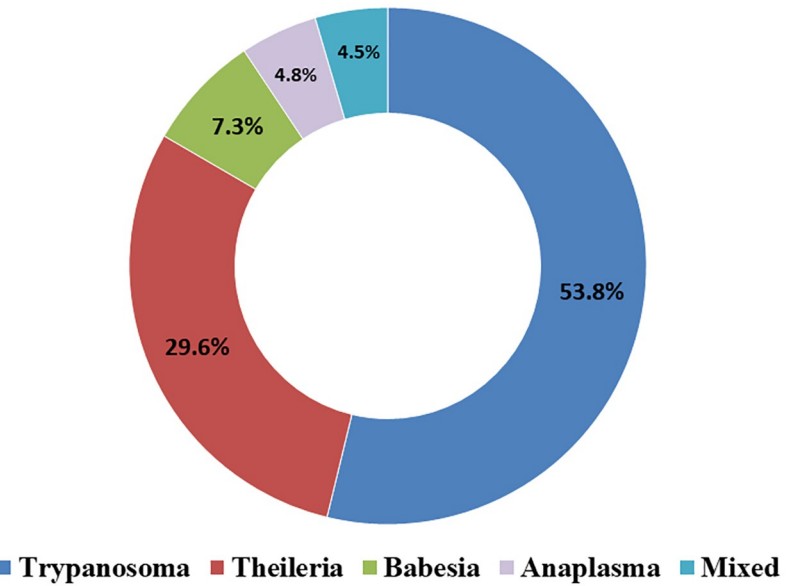

**Fig 2. The proportion of mixed infection with blood parasites among positive blood smears.**

reported during spring and autumn. The current study is considered the first study in Mosul city that examined the seasonality of blood parasites in animals. Our data for two successive years enabled us to better examine the seasonality of the reported blood parasites.

In this study, 61.77% of blood smears tested positive for blood parasite. This percentage is considered relatively high and indicates that the infection rate of blood parasites in animals in Mosul is substantial. The most prevalent parasite reported here was *Trypanosoma* spp. For the first time in Mosul, *Trypanosoma vivax* was diagnosed by Rhaymah and AL-Badrani [20] in

**Table 2. The proportion of blood parasites infection according to the age of the animal and the conditional logistic regression for the odds of the infection in adults compared to young animals.**

| Parasite | Age | Tested Animals | | Proportion of infection | OR | 95% CI | *P* |
|---|---|---|---|---|---|---|
| | Positive N = 78 | Negative N = 474 | | | | |
| **Trypanosoma** | | | | | | |
| Young | 35 | 43 | 44.87% | 1.00 | Referent[1] | NA |
| Adult | 163 | 311 | 34.39% | 0.65 | 0.39, 1.08 | 0.09 |
| **Theileria** | | | | | | |
| Young | 19 | 59 | 24.36% | 1.00 | Referent[1] | NA |
| Adult | 100 | 374 | 21.10% | 0.94 | 0.53, 1.69 | 0.84 |
| **Babesia** | | | | | | |
| Young | 2 | 76 | 2.56% | 1.00 | Referent[1] | NA |
| Adult | 27 | 447 | 5.70% | 1.24 | 0.27, 5.77 | 0.78 |
| **Anaplasma** * | | | | | | |
| Young | 2 | 76 | 2.56% | 1.00 | Referent[1] | NA |
| Adult | 18 | 456 | 3.80% | 2.54 | 0.58, 11.19 | 0.22 |

* Cattle only.

[1] Referent: a category of comparison for the other category.

Abbreviations: (OR): odds ratio, (CI): confidence interval.

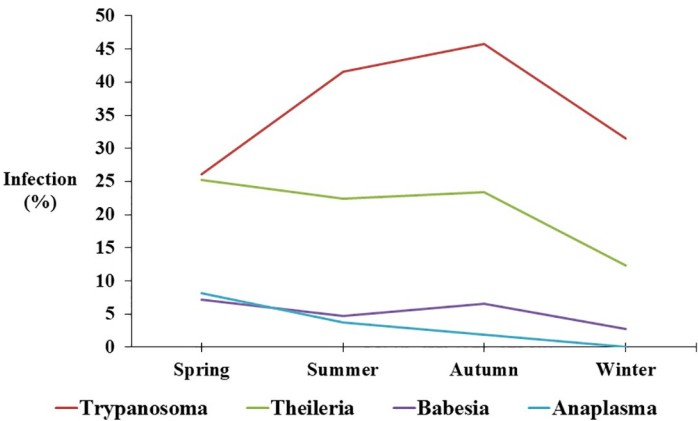

**Fig 3. Seasonality of blood parasites detected in blood smears.**

blood smears from imported calves, *Trypanosoma congolense* was identified by Hasan [21] in cattle and sheep, *Trypanosoma brucei* was detected by Al-Badrani [15] in cattle. Subsequently, *Trypanosoma* spp. was diagnosed in 6.5% of positive blood smears of cattle in Mosul, as the fourth most frequent blood parasite [2]. It seems that this infection has emerged to the city around 2011 as a result of uncontrolled importation [15, 20, 21], and then widely spread between local animals to become the most frequent blood parasite infection diagnosed in blood smears in the current investigation. More epidemiological research is important to identify its vectors, sources of infection, and risk factors in the city.

The study reported here revealed that *Theileria* spp. is a common blood parasite in local animals. The parasite has been diagnosed in several previous local studies in different animals. For instance, *Theileria hirci* was identified by Al-Obaidi and Alsaad [4] and Al-Obaidi [5] in local sheep and goats, respectively. In addition, *Theileria camelensis* was detected by Al-Saad et al. [22] in camels, and *Theileria annulata* was revealed by Alsaad et al. [14] in blood smears from newborn calves. Therefore, *Theileria* spp. is considered an endemic parasite in the area, and a percentage of about 35% among positive blood smears is considered high. Studies about the prevalence of *Theileria* spp. in animals in Mosul are very limited. The only available percentage of *Theileria* spp. infection among other blood parasites in cattle is 15.78% [2], which is lower than that reported in our study here, reflecting poor or ineffective control measures against the parasite and its vector.

In this study, *Babesia* spp. was reported at 8.4% among other blood parasites. This parasite has been identified in several previous studies, such as *Babesia equi* and *Babesia caballi* in horses [6, 23], *Babesia motasi*, *Babesia ovis*, *B. motasi*, *B. foliate*, and *B. taylori* in goats [14, 24], *Babesia bovis*, *B. bigemina*, and *B. divergens* in cattle [2, 25], and *Babesia caballi* in camels [7].

**Table 3. Season-to-season ratio as a measure of seasonality of the infection percentage according to the type of blood parasite.**

| Season-to-season ratio | Trypanosoma | Theileria | Babesia | Anaplasma |
|---|---|---|---|---|
| Spring to summer ratio | 0.83 | 1.49 | 2.05 | 2.88 |
| Summer to autumn ratio | 0.45 | 0.48 | 0.36 | 1 |
| Autumn to winter ratio | 2.14 | 2.80 | 3.52 | NA |
| Winter to spring ratio | 1.23 | 0.49 | 0.39 | NA |

Studies about the prevalence of *Babesia* spp. in animals in Mosul are also very limited. The only available percentage of *Babesia* spp. infection among other blood parasites in cattle is 10% [2], which is higher than that reported in our study here. It is difficult to consider that the prevalence of *Babesia* spp. in animals in Mosul city has decreased because the available prevalence by Al-Abadi and Al-Badrani [2] was for cattle only. More epidemiological surveys are required to confirm this finding.

The study conducted here revealed that *Anaplasma* spp. was diagnosed in blood smears from cattle only. In Mosul, however, *Anaplasma marginale* has been identified in buffalo [26], camel [27], while *Anaplasma ovis* was detected in goats [5, 14], in addition to that reported in cattle [3, 28]. Nevertheless, no previous study has reported the percentage of infection among other blood parasites except that for Al-Abadi and Al-Badrani [2] in cattle at 4.4%, which is lower than that reported in the study conducted here. An increase in the percentage of the infection is probably due to the absence of an effective control program against the parasite and its vector. Additionally, careful examination to blood smears from animals other than cattle is important as it was previously reported in goats, buffalo, and camels.

Results of the current data revealed that mixed infection is evident. This result is in line with the study of Al-Abadi and Al-Badrani [2] who reported mixed infection of blood parasites in cattle in Mosul. The presence of mixed infection with blood parasites is not unusual and is reported by other studies [29, 30]. On the other hand, our result showed no difference in the odds of the infection in adult animals compared to young. The current study cited eighteen local studies examined blood parasites in different animals [2–7, 14, 15, 20–28], however; none of them have reported the difference in the percentage of the infection among adults compared to young. Vieira et al. [30] did not show a difference in the percentage of the animals infected with *Babesia* spp. between different age groups, in contrast to that for *Anaplasma marginale*. It seems that the vector can infect adults as same as young animals; therefore, the odds of the infection are potentially not different.

The study reported here indicated seasonal differences in the infection. Our study is considered the first local study that examined the seasonality of the infection with blood parasites among animals. The seasonality of blood parasite infection in animals is evident as a function of the seasonal prevalence of the vectors [8]. Our results are in line with the study of Nnko et al. [9] who indicated that the infection with *Trypanosoma* spp. increases after the peak of the dry season by about three months. On the other hand, our study showed that the infection with *Theileria* spp., *Babesia* spp. increased during spring and autumn, which is in line with studies of, Asmaa et al. [11], Ayadi et al. [12], and Lempereur et al. [13]. Moreover, season-to-season ratio that took into account the difference in the length of the season confirmed the pattern of the seasonality observed by the percentage. Finally, additional studies are required to confirm these findings in local animals.

## Conclusions

The study concluded that the infection with blood parasites in different animals in Mosul is common with relatively high infection rate, the effect of age-related infection is negligible, and the seasonality of the infection is evident. Missing season-related investigations in the previous years could contribute to the endemic situation of the infection in the city. More epidemiological studies are required in order to outline control strategies against such infections.

## Acknowledgments

The authors thank the College of Veterinary Medicine, University of Mosul, for supporting this work.

## Author Contributions

**Conceptualization:** Mohammad O. Dahl.

**Data curation:** Hussam M. S. Alimam, Mohammad O. Dahl.

**Formal analysis:** Mohammad O. Dahl.

**Investigation:** Hussam M. S. Alimam, Dhiyaa A. Moosa, Eva A. Ajaj, Mohammad O. Dahl, Israa A. Al-Robaiee, Semaa F. Hasab Allah, Zahraa M. Al-Jumaa, Eman D. Hadi.

**Methodology:** Hussam M. S. Alimam, Dhiyaa A. Moosa, Eva A. Ajaj, Israa A. Al-Robaiee, Zahraa M. Al-Jumaa, Eman D. Hadi.

**Project administration:** Mohammad O. Dahl.

**Software:** Mohammad O. Dahl.

**Supervision:** Mohammad O. Dahl.

**Validation:** Mohammad O. Dahl.

**Visualization:** Mohammad O. Dahl.

**Writing – original draft:** Mohammad O. Dahl.

**Writing – review & editing:** Mohammad O. Dahl.

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
