## [Decision Letter · Decision Letter 0]

17 Jan 2022

PONE-D-21-39858Proportion and Seasonality of Blood Parasites in Animals in Mosul using the Veterinary Teaching Hospital Lab DataPLOS ONE

Dear Dr. Dahl,

Thank you for submitting your manuscript to PLOS ONE. After careful consideration, we feel that it has merit but does not fully meet PLOS ONE’s publication criteria as it currently stands. Therefore, we invite you to submit a revised version of the manuscript that addresses the points raised during the review process.

Many thanks for submitting your manuscript to PLOS One

It was reviewed by two experts in the field, and they have recommended some modifications be made prior to acceptance

I therefore invite you to make these changes and to write a response to reviewers which will expedite revision upon resubmission

I wish you the best of luck with your modifications

Hope you are keeping safe and well in these difficult times

Thanks

Simon

We look forward to receiving your revised manuscript.

Kind regards,

Simon Clegg, PhD

Academic Editor

PLOS ONE

Journal Requirements:

Reviewers' comments:

Reviewer's Responses to Questions

**Comments to the Author**

1. Is the manuscript technically sound, and do the data support the conclusions?

Reviewer #1: Yes

Reviewer #2: Partly

2. Has the statistical analysis been performed appropriately and rigorously? 

Reviewer #1: Yes

Reviewer #2: Yes

3. Have the authors made all data underlying the findings in their manuscript fully available?

Reviewer #1: Yes

Reviewer #2: Yes

4. Is the manuscript presented in an intelligible fashion and written in standard English?

Reviewer #1: Yes

Reviewer #2: No

5. Review Comments to the Author

Reviewer #1: This is an interesting epidemiological survey that provides new insights into the seasonality of infections caused by blood-borne pathogens in the animals of Mosul. I only have some minor suggestions for improvement, which are below, but other than that, I think it largely reads well!

1. In general, I would suggest changing "blood parasites" to "blood pathogens" or something that covers bacteria as Anaplasma, Ehrilichia, etc are not specifically parasites, even though some consider them as such.

I would also italicize any "et al" when referencing authors.

2. In your title, it may beneficial to specify what animals you examined, maybe "...livestock and dogs in Mosul..."?

3. In line 45 of your introduction, it should say "blood parasitic (or pathogenic if you change it) infections constituted..." and in line 68, it should say "such as those by Al-Obaidi...".

4. In your methods, I would recommend mentioning what animals you examined or refer to Table 1.

In line 90, it should be "tick infestations" and in your Laboratory Examination paragraph, it should read "Trypanosoma spp. were intercellular..." and the same for the other pathogens. In line 109-110, it should say "...and either single or mixed...".

I am also just curious, was there a reason the abundance (or number) of parasites/bacteria in each blood smear was not examined?

5. In your results, I would add the total number of samples examined between "...341 blood smears" and "(61.77%)...".

Line 143-144 should say "Additionally, the differences in the proportion...".

I would also only have "...at the Veterinary Teaching Hospital | University of Mosul between Oct 30, 2017 to Dec 31, 2019" on the first caption rather than each one.

Also can you explain what "Referent" means under 95% CI in Table 2?

6. In your discussion, line 198 should read "...city to examine..." and line 201 should say "...blood parasites".

In line 202, I would maybe change "burden" to "infection rate" as that sounds more like abundance.

In line 229-230, I would change it to read "[6, 23], Babesia ovis, B. motasi, B. foliate, and B. taylori in goats [14, 25]...".

Line 239-241 would be better as "...Anaplasma marginale has been identified in buffalo [27], camel [28] while Anaplasma ovis was detected in goats [5, 14] in addition to that reported in cattle...".

There should be an author referenced in line 242 and 249 after "for [2]" and "of [2]' and some references in line 247 after "...camels".

Line 252 should be "adult animals..." and in line 253, it should be "On the other hand, nineteen..."

Line 254 should read "...animals [reference]; however,..." and line 258, it should be "...adults as well as young...".

7. In your conclusion, I would again change "...substantial burden" to something like "a relatively high infection rate" in line 274.

Line 276-277 should be "...endemic understanding of the infection..."

8. Your tables and figures mostly look good but in Figure 3, Theileria is misspelled and I wonder if it also needs standard error/deviation bars?

I hope these suggestions help make minor improvements but it is a very fascinating paper otherwise!

Reviewer #2: Summary of the research:

I have summarised that the main research question for this journal entry is whether the time of year directly influences parasitic burdens within animals (cattle, sheep, goats, buffalo, horses, and dogs) that have been previously admitted to a Veterinary teaching hospital in Mosul, Iraq. This study also selected to focus on whether there was an association between age, seasonality, and parasitic burden. The research focuses on the detection of blood parasites (Trypanosoma spp., Theileria spp., Babesia spp. and Anaplasma spp., this also includes mixed infections) from Giemsa-stained smears and microscopy which remains the ‘Gold Standard’ tool of identification for blood parasite, and I therefore believe is accurately researched and reported. Within the study, the examination dates were the dates used to assign seasonality to each sample. It is stated that prior studies from Mosul involving blood parasites have been undertaken, however, an association with seasonality has been previously overlooked and is thus a gap in knowledge.

What are the claims and conclusions made in this study?

This study claims there is a relationship between seasonality and parasitic infections based on the season-to-season ratio adapted from Ferreira et al. The literature claims that Trypanosoma, Theileria and Babesia all peak in the Autumn-Winter ratio. They claim that Anaplasma has only been found predominantly within cattle samples, and this infection peaks in the Spring to Summer ratio, but there is no found evidence of the parasite occurring from Autumn to Winter or Winter to Spring ratios. It was also claimed that dogs predominantly harboured Babesia spp. They concluded that there is no significance between parasitic burden and age.

What is the relevancy, and does it fit with the existing literature?

I believe that the investigation, monitoring and understanding of animal parasites are a key part of improving veterinary care, medicine, and infection control measures; this can be used for preventative measures, emergency outbreaks, it can aid how animal husbandry may need to be altered to protect animal welfare and likewise, humans. It does fit with existing literature, since there is already published reports regarding the parasitic burden within Mosul, this piece of literature fills a gap in knowledge surrounding a correlation with seasonality which is an essential factor of the prevalence of parasites dependant on their life cycles, their environment and the animals present in these environments; this can aid/drive prevention and future parasite control in Mosul.

Minor Issues:

• Formatting issues (figures do not line up with the formatting e.g. Page 8, Fig 1. Page 9, Fig 2 and Page 11, Fig 3.)

• There is no mention of n (total sample size) analysed within the study animals/data collection section.

• Page 12, line 209: ‘it seems that this infection has emerged to the city around 2011’, this statement has no reference or further information to support the claim being made.

• Page 11, line 198, this does not read correctly and has grammatical issues.

• Page 12, line 201 ‘relatively considered high’, requires rewording.

• Page 14, line 235 ‘in Mosul in animal has’, grammatical issue.

• Page 15, line 260 ‘examined’ requires grammatical correction.

• Page 11, line 179 ‘increase in the in’ grammatical issue.

• Page 10, line 170 ‘the proportion of blood parasites infection’

Major Issues:

• Criterion of examining certain blood samples only from symptomatic (icteric mucous membranes, enlargement of superficial lymph nodes, haemoglobinuria, ticks, and prolonged illness) animals, has potentially limited the true reflection of parasitic infection prevalence within Mosul; therefore also limiting the true reflection of seasonal parasitic infections as only clinically unwell animals are examined microscopically due to circumstances of being brought to the Veterinary Hospital and blood taken under these animals meeting these conditions. Low burden parasitic infections can be asymptomatic, and equally, parasites are still found in clinically healthy animals, including cattle (example: https://parasitesandvectors.biomedcentral.com/articles/10.1186/s13071-016-1498-1).

• Total sample size is 552 animals. A larger sample size may have benefited this study and increased reliability of results and therefore the conclusions that can be drawn from a larger data set, by including animals that do not visually manifest a certain set of criteria. This can eliminate bias (only including animals manifesting certain symptoms) and more asymptomatic cases and low parasitic burdened animals could be found and may offer more data.

• The data used to ‘assign’ seasonality (Winter: 1st December to 28th February. Spring: 1st March to 31st May. Summer: 1st June to 30th September and Autumn: 1st October to 30th November) was stated as the examination date, not the sample collection date. This has the potential to be inaccurate due to possible delayed examinations and other circumstances that can delay the process of identification; unless otherwise stated that the examination date and collection date are equivalents, and do not differ. Is the examination date the same as the collection date of the original sample?

Additional comments:

A thoroughly intriguing study and I enjoyed reading it. This research fills an important gap within knowledge concerning parasitic burden in animals within Mosul, and it is a key element that can not be overlooked for animal and human health, with good reason. I would be happy to review this paper again.

6. PLOS authors have the option to publish the peer review history of their article (what does this mean?). If published, this will include your full peer review and any attached files.

Reviewer #1: No

Reviewer #2: No

---

## [Author Response · Author response to Decision Letter 0]

31 Jan 2022

The authors thank the reviewer for the review and critique of this manuscript. As a result, the authors have produced a better manuscript. A response to each point raised by the reviewers is included in the attached file.

---

## [Editor Report · Decision Letter 1]

4 Feb 2022

Proportion and Seasonality of Blood Parasites in Animals in Mosul using the Veterinary Teaching Hospital Lab Data

PONE-D-21-39858R1

Dear Dr. Dahl,

We’re pleased to inform you that your manuscript has been judged scientifically suitable for publication and will be formally accepted for publication once it meets all outstanding technical requirements.

Kind regards,

Simon Clegg, PhD

Academic Editor

PLOS ONE

Additional Editor Comments:

Many thanks for resubmitting your manuscript to PLOS One

As you have addressed all the comments and the manuscript reads well, I have recommended it for publication

You should hear from the Editorial Office shortly.

It was a pleasure working with you and I wish you the best of luck for your future research

Hope you are keeping safe and well in these difficult times

Thanks

Simon

---

## [Editor Report · Acceptance letter]

9 Feb 2022

PONE-D-21-39858R1 

Proportion and Seasonality of Blood Parasites in Animals in Mosul using the Veterinary Teaching Hospital Lab Data 

Dear Dr. Dahl:

I'm pleased to inform you that your manuscript has been deemed suitable for publication in PLOS ONE. Congratulations! Your manuscript is now with our production department. 

Kind regards, 

on behalf of

Dr. Simon Clegg 

Academic Editor

PLOS ONE